# "Silver" Generation at Work—Implications for Sustainable Human Capital Management in the Industry 5.0 Era

Agnieszka Laskowska * 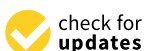 and Jan Franciszek Laskowski 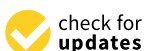

Faculty of Management, Lublin University of Technology, 20-618 Lublin, Poland
* Correspondence: a.laskowska@pollub.pl

**Abstract:** The demographic low, and thus the inevitable aging of the populations of economically developed countries, makes it necessary to extend the working life of citizens. Therefore, an increasing percentage of workers are representatives of the so-called "silver" generation, i.e., people over 50 years of age. The main objective of the study is to characterize the system of values, attitudes to work, and related expectations of professionally active women and men over 50 years of age and to determine whether, and if so, how the hierarchy of values, attitudes to work, and job satisfaction affect the further development of their professional careers. In order to achieve such a goal, a CAVI survey was conducted on a sample of $n = 600$ respondents. The statistical significance of the differences between the groups was tested with the Mann–Whitney U test. To verify the main hypothesis, ordinal logistic regression models were built. The results of the study were supplemented with elements of descriptive statistics. In the course of the research work, it was confirmed that there are significant differences between the studied men and women in the assessment of traits concerning their value hierarchy, attitudes towards work, and career prospects. Traits such as value hierarchy and attitudes toward work have a significant impact on the career development of both women and men of the "silver" generation, while job satisfaction shows a significant impact on career development only for women.

**Keywords:** "silver" generation; multigenerational teams; sustainable human capital management

## 1. Introduction

Due to the high dynamics and unpredictability of changes taking place in the modern economy and the constant increase in competition on the market, the ability to attract and retain experienced and talented employees plays a key role in obtaining a sustainable competitive advantage of the company.

The demographic decline, and thus the inevitable aging of societies in economically developed countries (EU, USA, Japan), and a significant improvement in the living conditions and health of citizens of these countries, with the simultaneous dynamic economic growth generating an oversupply of jobs, makes it necessary to extend the period of their professional activity. According to the data of the Central Statistical Office, in 2020 the percentage of people over 60 in Poland reached the level of 25.3%, while in 2050 these people will constitute about 40% of the total [1]. A similar trend can be seen across the EU, where it is estimated that one-third of the population will be over 65 by 2060 [2]. As a consequence, an increasing percentage of employees currently functioning on the labor market are representatives of the so-called "silver" generation, that is people over 50 years of age. These employees differ significantly from their younger colleagues (generation Y and Z), starting from their value system, approach to work, and professional career, to expectations towards work and the employer. It should be noted that the attitudes and expectations of employees towards work and professional career are constantly changing, evolving under the influence of various social, economic, and political factors, which implies the need to

identify these factors and the relationships between them, as well as to determine their development tendencies.

This problem appears to be particularly relevant as the world enters the Industry 5.0 era and the consequent economic and social changes we are currently undergoing. The Industry 5.0 concept complements the existing Industry 4.0 approach, where through the development of innovative technologies, industrial processes, supply chains, and new business models, the transition to a sustainable, human-centered, and resilient industry is sought. Industry 5.0 provides a vision of over-engineered businesses that go beyond efficiency and productivity understood as primary goals. The concept places the worker at the center of the manufacturing process and leverages new technologies to deliver prosperity beyond jobs and growth while respecting the planet's productive limits [3].

All of this seems to be particularly important in the context of the unfavorable demographic changes currently being experienced by the societies of economically developed countries. This implies the need to identify the beliefs, needs, and expectations of workers over the age of 50, who will soon become the main driving force of the economy.

The results of this study will allow employers to more effectively activate employees of the "silver" generation (50+), and thus to fully use the potential of multigenerational teams. The result will be a more sustainable, optimal, effective, and innovative management of the human capital of enterprises in the era of Industry 5.0.

There are several definitions of the term "generation" in the literature on generational differences. For the first time, this term was described by Mannheim in 1928 in the work entitled *Das Problem der Generationen*. According to the researcher, a "generation" is a cohort of people of similar age who experience common historical events. The inherent feature of this social construction is the common awareness of the experienced fate, similar attitudes and behaviors, goals, value systems, and principles of operation and interpretation of reality [4]. A similar definition was repeated by Ryder in 1965, who described a "generation" in more detail as "a group of individuals who experienced the same event over the same time period" [5]. Contemporary definitions of "generations" do not differ much from those quoted above. For example, Kupperschmidt defines a "generation" as "an identifiable group united by years of birth, age, location and significant life events in critical stages of development" [6]. In summary, for the purposes of this study, a "generation" is defined as "a group of people roughly the same age who experience and are influenced by the same set of important historical events at key periods in their lives, usually in late childhood, adolescence and early adulthood".

Based on the analysis of the literature on generational issues at work, five main groups of employees can be distinguished: "Silent Generation" (1922–1944), "Baby Boomers" (1945–1964), "Generation X" (1965–1980), "Generation Y"—so-called millennials (1980–1994), and "Generation Z" (born after 1994) [7–15]. It is important to note that while most authors have adopted a common nomenclature and similar general timeframes to define particular groups, there is considerable discrepancy in the literature as to when exactly each generation begins and ends [16].

In the light of demographic changes currently taking place on the labor market, employers are forced to manage the organization with particular emphasis on the generational diversity of employees [17]. The problem of managing multigenerational teams is widely analyzed in the literature [7,12,18–21]. It is one of the most important parts of the issue of sustainable management of a company's social [22] and economic capital [23–25]. Research on generations in the labor market usually focuses on distinguishing differences between individual generations [15,18,21], as well as on defining a catalog of features characterizing employees currently functioning in the labor market [8,26–29]. Currently, there are many companies employing three, and sometimes even four generations of employees, often with different preferences, values, and work styles. Therefore, it is very important in the context of proper management of such a team to thoroughly diagnose the characteristics of employees from particular groups [30,31]. At the same time, special attention should be paid to the skillful separation of characteristics resulting from generational affiliation

from those resulting from other factors, such as organizational experience, seniority, and technological progress, which according to many scientists, is the main methodological challenge in generational research [32,33].

The impact of the systematic aging of the population is now an increasing problem for dynamically developing labor markets. Indeed, the age structure of the European population is expected to change significantly over the next decades. By 2060, the share of people over 65 years of age will increase from 18% to 30% compared to now, while the share of people over 80 years of age will be more than double. At the same time, the percentage share of the working age people (15–64 years) in the total population is expected to substantially decrease, from 67% to 57% [34]. In Poland, also, tendencies to increase the number and share of elderly people in the total population can be observed. At the end of 2020, there were 14.4 million people aged 50 and over. They accounted for 37.6% of the total population and this share increased by 0.2% in relation to the previous year [35]. It is estimated that in 2050 the percentage of people over 60 in Poland will reach 40% of the total population [1]. Therefore, to maintain production and service capacities, companies are increasingly trying to encourage older workers to remain professionally active for longer. In order to achieve this, it is necessary to remove technological [36] and social barriers that make it difficult to continue working and, most importantly, to learn about expectations and build an effective incentive system based on them, which will encourage professionally experienced people to postpone the decision to retire [20,34].

The need to prolong professional activity is also emphasized in the scientific literature, where the term "silver generation" was generally defined as a group of active people aged over 50–55 [34,37–39]. Taking into account the previously cited taxonomy of generational groups, people who are currently (2022) 50–55 years old were born after 1967, i.e., they belong to the so-called Generation X.

When reviewing the literature related to the topic of generations in the labor market, several basic groups of characteristics can be distinguished. Most often, generational cohorts are defined by the professed system of values, their attitude towards work, the level of professional satisfaction, and expectations regarding career development [7,9,11,14,27,40–43]. Therefore, in the course of the research work, such a division of features characterizing employees of the "silver" generation was adopted.

The main purpose of this work is to characterize the system of values, approach to work, and the related expectations of professionally active women and men over 50, and to determine whether, and if so, how the hierarchy of values, attitudes towards work, and job satisfaction affect the further development of their professional careers. The analysis of the literature on the subject, carried out at the initial stage of the investigation, allowed for the formulation of a general hypothesis ($H_G$), which states that:

> *Characteristics such as the hierarchy of values, attitudes towards work, and professional satisfaction of both men and women employees of the "silver" generation have a significant, positive impact on the further development of their professional careers.*

Each of us has values that are very important to us, perhaps completely irrelevant to someone else. Some of them are preferred and highly valued by humans, creating an individual hierarchy of values. The term "value", "being valuable", entered the vocabulary at the end of the 19th century. Currently, the *Oxford* English *Dictionary* defines the concept of "value" as "the regard that something is held to deserve; the importance, worth, or usefulness of something" or "principles or standards of behavior; one's judgment of what is important in life" [44]. In the literature, a number of theories characterize catalogs of human characteristics, which are also descriptors of his system of values [45–49]. The common part of these works is the five-element characterization of values, which says that [50]:

(1)　Values are concepts or beliefs that are;
(2)　Relate to the desired goals, describing the final states of affairs or behavior;
(3)　Transcend specific situations;

(4) Guide the selection and evaluation of behaviors and events;
(5) Are ordered by importance.

On the basis of these assumptions, the currently most popular theory of basic human values was created, the author of which is Shalom H. Schwartz. The main thesis of this theory is the assumption that the structure of human values is in the shape of a universal, motivational, circular continuum [51–53]. In his studies, Schwartz also describes two rules ordering the values in the circle (Figure 1), i.e., the rules of compatibility and conflict. The first of these rules states that values adjacent to each other in a circular model of values are possible to implement together because they are a cognitive representation of similar goals. On the other hand, the second rule assumes that values lying on opposite sides of the circle are not possible to realize together because they are cognitive representations of contradictory goals [50].

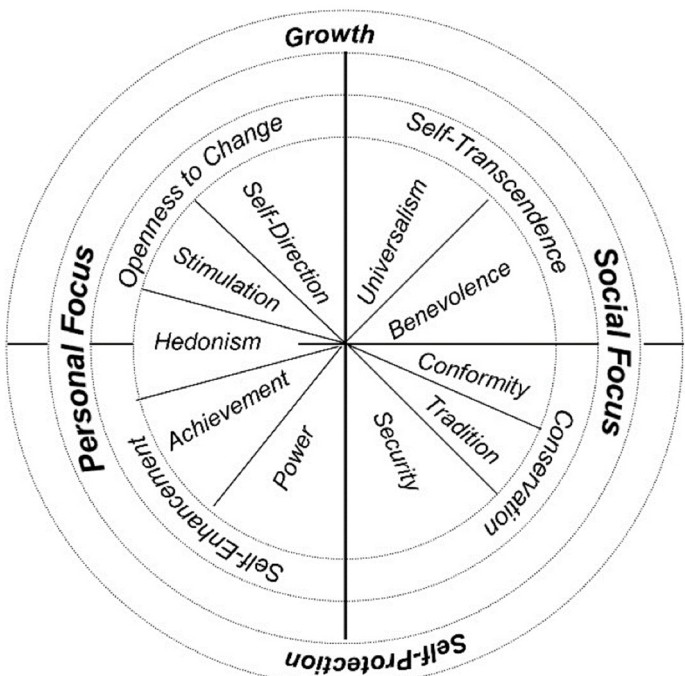

**Figure 1.** Schwartz model of human values. Own elaboration based on [52,54].

The closer a value is to another value in the circle, the more similar are the motivations it expresses. Contrastingly, the more distant two values are, the more antagonistic they are in their motivations. Such an arrangement allows viewing values as organized along with a higher order of two bipolar dimensions: opposing self-transcendence (benevolence and universalism) to self-enhancement (power and achievement) values, and opposing conservation (security, conformity, and tradition) to openness to change (self-direction, stimulation, and hedonism) values [54].

Previous research on values in human life proves that they are strictly dependent on the age of the respondent and change over time [55]. People's value systems differ depending on the country and cultural group they come from [51,56]. This diversity results from different historical and contextual circumstances, including different levels of political and economic development in the surveyed societies [57]. Interestingly, there is also a very strong differentiation in the priorities of women's and men's values [58].

According to these assumptions, a research hypothesis ($H_1$) can be formulated, which says that there are significant differences between the value systems of women and men representing employees of the "silver" generation.

Generally speaking, the term "work" is defined as "activity involving mental or physical effort done in order to achieve a purpose or result" [44]. In other words, it is any

purposeful activity leading to the satisfaction of human needs that has social inclinations. Taking into account the values that work brings to an individual, three types of attitudes toward it can be distinguished: autotelic attitude, occurring when work is an end in itself and a way to meet the need for self-development; instrumental attitude, consisting of treating work as a means to the satisfaction of the basic needs of the employee, such as security and social contacts; and a punitive attitude, characterized by seeing work as a punishment, something unpleasant that must be avoided [59]. To these attitudes, one can add a patriotic attitude, characterized by perceiving work as a service to society and fulfilling one's duty to the homeland. It should be noted that contemporary studies of attitudes towards work do not focus on the attitude of a person to work as such, but consider the attitudes of an individual related to a specific, currently performed job. In this case, the following can be used as measures of attitudes towards work: the level of job satisfaction, commitment to work, and the sense of importance of work [59,60]. Research on people's attitudes towards work shows that, as in the case of a value system, they depend on age and change over time. Since age is usually treated as a determinant of generational affiliation, it is commonly assumed that the attitudes towards work of representatives of different generations are different [16,40,61,62]. Furthermore, the conducted research also showed differences in attitudes towards work in terms of gender [63–65].

Based on these assumptions, a research hypothesis (H$_2$) can be formulated, which states that there are significant differences between the attitudes towards work of women and men representing employees of the "silver" generation.

Another very important descriptor of the "silver" generation employees is job satisfaction. One of the most frequently cited definitions of job satisfaction in the literature defines it as "a pleasant emotional state resulting from an individual's perception of his or her job as realizing or giving an opportunity to realize significant values available at work, provided that these values are consistent with their needs" [66]. Researchers dealing with the issue of job satisfaction point to its two basic aspects: emotional (affective) and cognitive [67]. The affective factor consists of the employee's feelings about work (short-term or permanent attitudes towards work), while the cognitive factor consists of what the employee thinks about the job (judgments about the work environment and tasks performed). This set is sometimes supplemented with a third factor, which is subjectivity in the perception of the situation resulting from individual characteristics (age, gender, nationality, and life experiences). The main factors determining job satisfaction are personal factors (needs, age, sex, and experience), organizational factors (type of work, remuneration, promotion opportunities, and working conditions), and social factors (organizational culture, ethics, relations with the supervisor, and cooperation in team) [68]. The level of employee satisfaction can have a significant impact on customer satisfaction and loyalty [69]. The most commonly used measures of job satisfaction are: remuneration, development opportunities, general evaluation of work, teamwork, organization and management, and working conditions [70–72]. Research on the differences in professional satisfaction depending on gender proves that there are significant differences between men and women in this aspect [73–79]. Previous research also suggests that job satisfaction is not constant and changes over time. Some studies show a tendency to decrease job satisfaction in the 40–50 age group, with job satisfaction steadily increasing up to this point [80].

According to these assumptions, a research hypothesis (H$_3$) can be formulated, which states that:

> *There are significant differences between the level of professional satisfaction of women and men representing employees of the "silver" generation.*

The concept of career development was first introduced to the literature in 1951 by Ginzberg, Ginsburg, Axelrad, and Herman. The basis for their considerations was the assumption that the choice of profession is a developmental process that lasts many years and ends in early adulthood [81]. In the following years, Ginzberg clarified his position, assuming that it is a process of making professional decisions (professional choices) throughout life. This view is close to career development theorists [82]. Brown and Brooks described

career development as "the process of preparing for a choice, a process of choosing and constantly making choices from the many professions available in society" [83]. Edgar Schein presented an interesting concept of career development. Assuming that there is a relationship between the values of a given person and the type of career chosen by him, he created the concept of "career anchors", which are associated with directing a given person to one of eight possible values: professionalism, management (leadership), autonomy and independence, security and stability, creativity and entrepreneurship, idealism (meaning, truth, and dedication to others), challenges, and lifestyle [84,85]. In turn, John Holland in his research proved that people are looking for a work environment that will allow them to achieve self-realization, i.e., one that is consistent with their skills, personality, and preferences [86]. Currently, we are seeing a gradual disappearance of the traditional linear career model, associated with stable working conditions and predictability of a professional career, towards a multidirectional, dynamic, and fluid career path. Examples of currently pursued careers can be protean career, borderless career, or kaleidoscope career. A protean career is associated with activity and a focus on development. A person pursuing such a career is guided by their own value system when making decisions and actions, is proactive and independent, and is focused on development [87]. A borderless career is associated with mobility, moving from one company to another, and it may also concern the use of all opportunities to develop one's competencies and professional potential in a given workplace. This career model can also be associated with the disappearance of boundaries between professional activity and other spheres of life. Such a career can involve crossing barriers related to physical mobility related to a change of industry or employer, and psychological mobility meaning readiness to change career [88]. The concept of a kaleidoscopic career (ABC) emphasizes the fact that a person pursuing a career must face three key issues important for their development: authenticity, balance, and challenges. These areas form the "mirrors" of a kaleidoscope that can be related to the way of pursuing a career, giving unlimited possibilities to create different and unique career patterns. This model shows the perspective of pursuing different career paths, which depends on personal choices, activities undertaken, and the way of reacting to difficulties [89].

The studies conducted so far on the differentiation of professional career development depending on gender prove that there are significant differences between men and women in this aspect [90–93]. According to these assumptions, a research hypothesis (H$_4$) can be formulated, which states that

> *There are significant differences between the expectations regarding the professional career development of women and men representing employees of the "silver" generation.*

## 2. Materials and Methods

### 2.1. Research Tools

The main research tool used to achieve the purpose of the work was the CAVI questionnaire. The survey was anonymous and included a sample of *n* = 600 Poles (377 women and 223 men), participants of the *Badanie Opinii* internet panel. This condition was necessary due to the selected sampling frame. The survey covered respondents aged 50 and more who are professionally active. More than half (52%) of the respondents had a secondary education. The largest percentage of respondents resided in large cities (23%) and were employed in commerce (27%) and industry and construction (15%). Detailed characteristics of the sample are presented in Table 1. The attempt was intentional. The study was conducted in the first quarter of 2022.

**Table 1.** Sociodemographic characteristics of the study sample.

|  |  | *n* | % |
|---|---|---|---|
| Sex | Woman | 377 | 62.83% |
|  | Man | 223 | 37.17% |
| Age | 50–55 years | 303 | 50.50% |
|  | 56–60 years | 149 | 24.83% |
|  | 61–65 years | 110 | 18.33% |
|  | 66 years and older | 38 | 6.33% |
| Education | Elementary | 5 | 0.83% |
|  | Basic vocational | 69 | 11.50% |
|  | Secondary | 314 | 52.33% |
|  | Higher Education | 212 | 35.33% |
| Residence | Village | 106 | 17.67% |
|  | City of up to 20,000 residents | 69 | 11.50% |
|  | City of 20,000 to 50,000 residents | 98 | 16.33% |
|  | City of 50,000 to 100,000 residents | 82 | 13.67% |
|  | City of 100,000 to 250,000 residents | 107 | 17.83% |
|  | City with more than 250,000 residents | 138 | 23.00% |
| Employment | Public administration | 53 | 8.83% |
|  | Middle and senior management personnel | 36 | 6.00% |
|  | Transportation | 30 | 5.00% |
|  | Industry and construction | 94 | 15.67% |
|  | Information technology (IT) | 15 | 2.50% |
|  | Education, higher education | 54 | 9.00% |
|  | Uniformed Services | 15 | 2.50% |
|  | Commerce | 167 | 27.83% |
|  | Energy | 13 | 2.17% |
|  | Health care | 46 | 7.67% |
|  | Banking and finance | 38 | 6.33% |
|  | Legal Services | 7 | 1.17% |
|  | Other | 32 | 5.33% |

The content part of the questionnaire included seven questions with a semantic differential scale, constructed in accordance with Charles E. Osgood's theory of semantic differences [94,95]. The scales used have values from 1 to 10, where 1 indicates the least significant values and 10 the most significant. The intervals between successive values of the scales were designed to be equal, making them interval scales. In order to conduct statistical analyses and the construction of ordinal logistic regression models, the questions were grouped into three blocks forming independent variables: I—value hierarchy (16 factors analyzed), II—attitudes toward work (3 factors analyzed), and III—job satisfaction (3 factors analyzed), and one block forming the dependent variable: IV—career development (17 analyzed factors). In order to make the research more precise, the survey questionnaire was also supplemented with five descriptive questions with an ordinal scale, which made it possible to determine the place that work occupies in the lives of representatives of the "silver" generation, their propensity to continue working after reaching retirement age, occupational mobility, and willingness to share their experience. The reliability of the survey questionnaire was checked by analyzing internal consistency using Cronbach's Alpha ($\alpha$) and McDonald's omega ($\omega$) tests. Both Cronbach's Alpha and McDonald's omega ($\omega$) reliability indexes proved satisfactory, being $\alpha = 0.72$–$0.91$ and $\omega = 0.81$–$0.90$, respectively. All calculated values are shown in Table 2.

**Table 2.** Results of the reliability analysis.

|  | Cronbach's Alpha ($\alpha$) | McDonald's Omega ($\omega$) |
|---|---|---|
| I. Hierarchy of values | 0.911 | 0.906 |
| II. Attitude towards work | 0.724 | 0.817 |
| III. Professional satisfaction | 0.833 | 0.838 |
| IV. Professional career development | 0.904 | 0.903 |

*2.2. Data Analysis*

The data collected in the study were subjected to a reliability analysis and then the questions were aggregated within the blocks. In each of the blocks, a test that compared the answers of women and men was performed. The normality of the distributions was checked with the Shapiro–Wilk test. The Mann–Whitney U test was performed to check the statistical significance of differences between the examined groups of women and men ($H_1$, $H_2$, $H_3$, and $H_4$). In order to verify the main hypothesis ($H_G$), models of ordinal logistic regression were built within the group of men and (separately) women. Logistic regression is a transformation of a linear regression using the sigmoid function. The vertical axis represents the probability of a given classification and the horizontal axis is the value of *x*. It assumes that the distribution of *y* and *x* is the Bernoulli distribution [96]. The ordinal logistic regression model is a generalized linear model and is characterized by the fact that the dependent variable has a finite number of levels. A proportional odds model was adopted (the probability of moving from each value to a higher one is proportional). The construction of two separate models for women and men results from the assumed statistical significance of differences between these groups.

Then, after the aggregation of questions in blocks, the dependent variable, i.e., block IV, was divided into classes, following the principle of building a distribution series. The minimum, maximum, and range values were found and then the length of the class was calculated, assuming that there would be 10 classes. Each class was assigned a rank from 1 to 10 (such grouping was used to make it possible to perform an ordinal logistic regression).

On the basis of the type I (LR 1) log-rank test, the significance of the variables entering the model in turn was determined. The log-rank I test checks whether a given variable significantly improves a model based solely on the intercept. Then, the goodness of fit of the model was examined, i.e., scaled $\chi2$ was calculated. The value of the scaled coefficient $\chi2$ should not exceed 1 or be very close to this value—this indicates that the data is not excessively dispersed. For values above 1, the dispersion should also be estimated. Based on the Pearson residuals plot, outliers were discarded. The significance of the independent variables was verified using the Wald test. Probabilities of assuming certain levels of the dependent variable were estimated. The results of the logistic regression were presented using the odds ratio, which allows determination of how many times the chance of a higher assessment of the phenomenon increases or decreases when the independent variable has increased by one unit. The research results were supplemented with elements of descriptive statistics.

**3. Results and Discussion**

In all analyses, the significance level of $p = 0.05$ was adopted.

*3.1. Differences between Groups*

After performing the reliability analysis, the questions were aggregated within the blocks, creating appropriate indexes for them. In each of the blocks, a test was performed that compared the responses of women and men (Table 3). The normality of distributions was checked with the Shapiro–Wilk test—in all groups $p < 0.0001$. Due to the lack of a normal distribution for the dependent variable within the groups, the Mann–Whitney U test was performed (Table 4). This test is a nonparametric equivalent of the t-test for two independent groups, which compares the rank sums in the two groups.

**Table 3.** Statistics for blocks I–IV.

| Overall | *n* | Mean | Def. Std. | Median | Min. | Max. |
|---|---|---|---|---|---|---|
| I. Hierarchy of values | 600 | 130.4 | 19.38 | 134 | 40 | 160 |
| II. Attitude towards work | 600 | 146.9 | 22.20 | 149 | 43 | 190 |
| III. Professional satisfaction | 600 | 24.3 | 5.57 | 25 | 4 | 35 |
| IV. Professional career development | 600 | 172.3 | 39.56 | 178 | 12 | 238 |
| **Men** | *n* | Mean | Def. Std. | Median | Min. | Max. |
| I. Hierarchy of values | 223 | 126.7 | 20.19 | 130 | 40 | 160 |
| II. Attitude towards work | 223 | 142.7 | 22.84 | 143 | 43 | 190 |
| III. Professional satisfaction | 223 | 24.0 | 5.69 | 25 | 4 | 35 |
| IV. Professional career development | 223 | 167.2 | 40.55 | 173 | 12 | 236 |
| **Women** | *n* | Mean | Def. Std. | Median | Min. | Max. |
| I. Hierarchy of values | 377 | 132.7 | 18.55 | 138 | 59 | 160 |
| II. Attitude towards work | 377 | 149.4 | 21.45 | 152 | 49 | 190 |
| III. Professional satisfaction | 377 | 24.5 | 5.49 | 25 | 6 | 35 |
| IV. Professional career development | 377 | 175.3 | 38.71 | 183 | 44 | 238 |

**Table 4.** Mann–Whitney U test results for each of the four blocks.

| | Sum of Ranks M | Sum of Ranks W | Z | *p* | Median M | Median W | *n* M | *n* W |
|---|---|---|---|---|---|---|---|---|
| I. Hierarchy of values | 58,472.5 | 121,827.5 | −4.16 | 0.00003 | 130 | 138 | 223 | 377 |
| II. Attitude towards work | 58,893.5 | 121,406.5 | −3.96 | 0.00008 | 143 | 152 | 223 | 377 |
| III. Professional satisfaction | 64,579 | 115,721 | −1.19 | 0.23594 | 25 | 25 | 223 | 377 |
| IV. Professional career development | 61,542 | 118,758 | −2.67 | 0.00769 | 173 | 183 | 223 | 377 |

3.1.1. The "Silver" Generation Hierarchy of Values

For the research hypothesis $H_1$ which says that there are significant differences between the value systems of women and men representing the employees of the "silver" generation, the null hypothesis that there are no differences between the groups was rejected in favor of the alternative hypothesis that the differences between the groups are significant (Z = −4.16; $p$ = 0.00003). The answers given by women differ significantly from the answers given by men. Women gave, on average, higher marks in questions concerning the hierarchy and the system of values.

To examine the system of values that the representatives of the "silver" generation follow in their lives, the respondents were asked to evaluate sixteen characteristics selected according to the assumptions of Shalom H. Schwartz's theory of basic human values [52] and assign each of them a score on a scale from 1 to 10, where 1 is the least important and 10 is the most important. These ratings are presented in Figure 2 as an arithmetic average of the responses obtained. Answering the question what values are most valued by you in life, the respondents gave the highest scores to values such as family (9.08), honesty (8.86), love (8.54), happiness (8.42) and health (9.03), security (9.01), compliance (8.74), and stabilization (8.68). It should be noted that the ratings given to these features by women differed significantly from the ratings given by men (Figure 2). Women gave higher marks on average by approx. 0.5 points. Differences also occurred in the hierarchy of importance of the above-mentioned characteristics, where women, unlike men, preferred happiness and security over love and health. The values rated lowest by respondents include admiration and respect (6.29), prosperity, wealth, and money (6.75), professional/social position (6.91), and work and career (7.5). Moreover, in this case, women gave higher marks than men (on average by approx. 0.2 points).

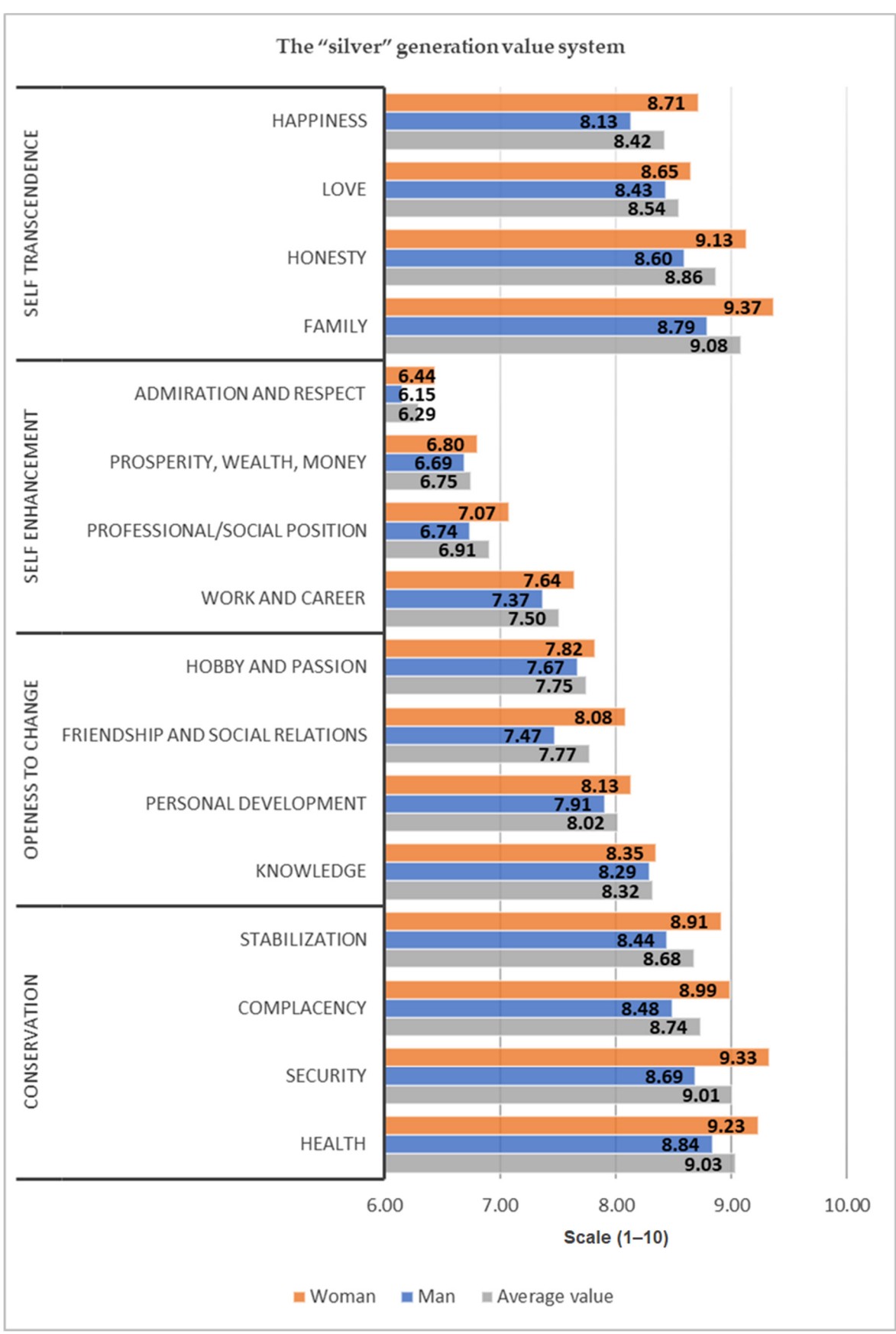

**Figure 2.** The "silver" generation value system.

In order to properly characterize the value system of the "silver" generation, the above-mentioned features, after appropriate grouping, were entered into the circular model of Shalom H. Schwartz's value structure (Figure 3). These ratings are presented in Figure 3 as an arithmetic average of the responses obtained. According to the assumptions of the model, four antagonistic groups of higher-order values were created, i.e., self-transcendence (family, honesty, love, and happiness), self-enhancement (work and career, professional/social position, prosperity, wealth, money, admiration, and respect), conservation (health, security, compliance, and stabilization), and openness to change (knowledge, personal development, friendship and social relations, and hobby and passion). The four above-mentioned groups of values also create two independent dimensions of the highest order of values (social focus—personal focus and growth—protection), and their opposite poles create oppositional values. Analyzing the scheme created in this way, it can be said that the surveyed representatives of the "silver" generation are people focused on others (family) more than on themselves (social focus), a rather conservative approach to life's challenges (conservation), valuing spiritual values more than material ones (self-transcendence). These people are usually conservatives who value personal security, health, stabilization, and compliance more than any type of dynamic change. They are reluctant to undertake activities that require them to be more active than the standard, engage, and devote their free time. They consider universal values such as family, honesty, love, and happiness to be the most important in life. Those values are placed much higher than work and career, professional/social position, prosperity, wealth, money, admiration, and respect, which corresponds to their conservatism.

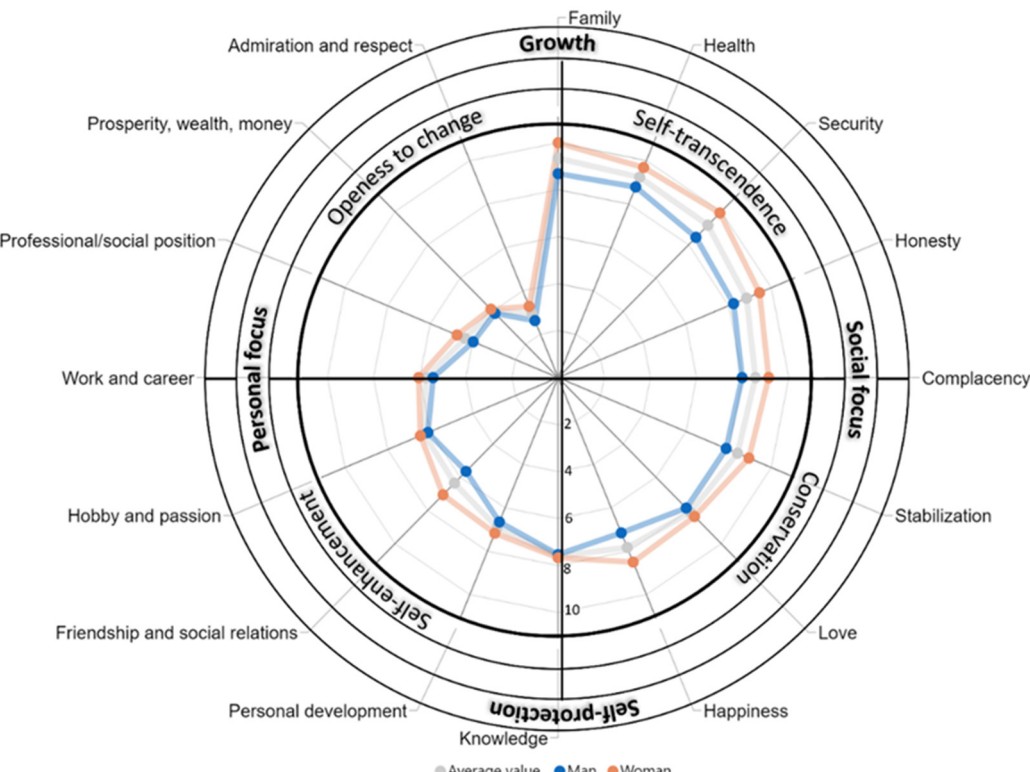

**Figure 3.** Schwartz model of the "Silver" generation value system.

It should be noted here that the value system of the "silver" generation presented above is mostly consistent with the characteristics of representatives of the "Generation X" available in the literature [97,98]. The differences here are in the relatively low assessment of the value of the self enhancement category, which may result from the relatively low level of job satisfaction characteristic of Poles, associated with progressive professional

burnout [99]. This phenomenon seems to confirm the theory that the system of values is strictly dependent on the age of the respondent and evolves over time [55].

### 3.1.2. Attitude towards Work of the "Silver" Generation

For the research hypothesis $H_2$ which says that:

*There are significant differences between the attitudes towards the work of women and men representing the employees of the "silver" generation.*

The null hypothesis that there are no differences between the groups was rejected in favor of the alternative hypothesis that the differences between the groups are significant ($Z = -3.96$; $p = 0.00008$). The answers given by women differ significantly from the answers given by men. Women gave, on average, higher marks in questions concerning the attitude and attitude towards work.

To examine the attitude towards work of the representatives of the "silver" generation, respondents were asked to evaluate three statements formulated according to the assumptions of the value theory that work brings to an individual [59] and to assign each of them scores on a scale of 1 to 10, where 1 represents the least significant values and 10 the most significant. These ratings are presented in Figure 4 as an arithmetic average of the responses obtained. According to the conducted research, the statement that best characterizes the respondents' attitude to work is: work is a source of prosperity, security, financial stability, and respect (8.09). At a slightly lower level (7.48) is the statement: work is a source of satisfaction, self-esteem, and motivation. It should be noted that in both cases, the ratings given by women differed from the ratings given by men (Figure 4). Women gave higher marks than men by 0.35 and 0.6 points, respectively. The lowest-rated statement by the respondents was: work is a source of stress and fatigue (5.29). Moreover, in this case, the scores awarded by women differed from those of men (they were lower by 0.29 points).

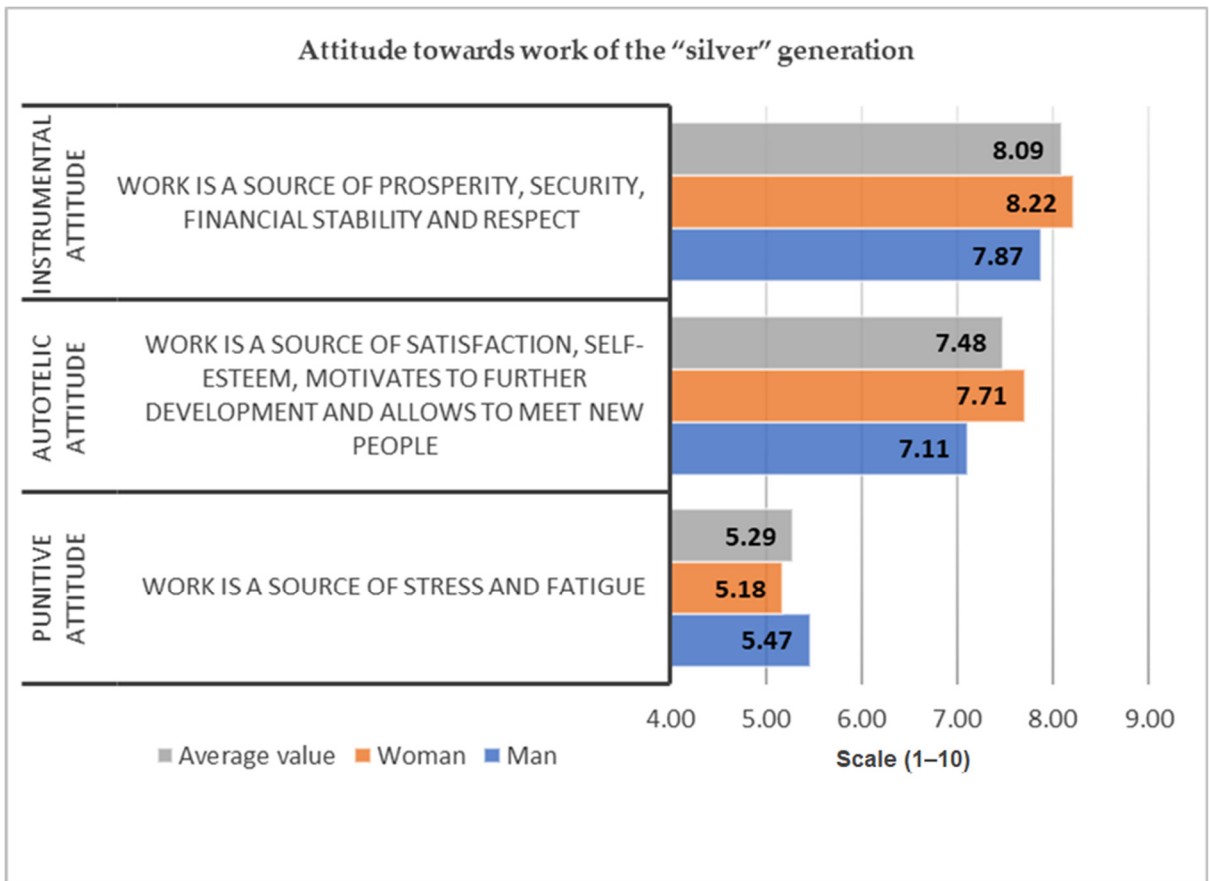

**Figure 4.** Attitude towards work of the "silver" generation.

In order to detail the characteristics presented above and determine the place of work in the lives of representatives of the "silver" generation, respondents were also asked to answer two questions about the balance between work and private life. In both cases, the responses given by women differed from those given by men. The ability to maintain a balance between private and professional life is important for 96% of women and 91% of men (Figure 5).

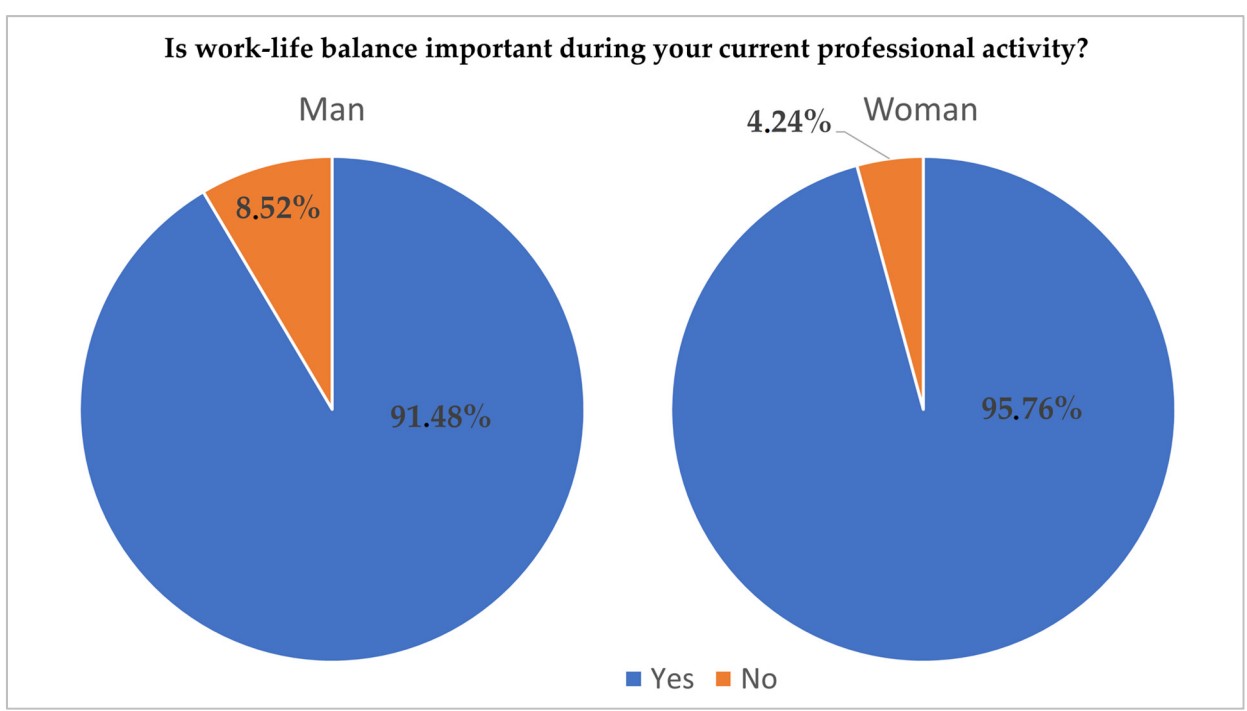

**Figure 5.** Work-life balance during current professional activity.

Interestingly, 31% of men believe that work is just as important as family and friends, while as many as 65% of them declared that family and friends are more important to them than work. In the case of women, 48% of them believe that work is just as important as family and friends, and almost 50% declared that family and friends are more important to them than work (Figure 6). This is probably due to the significantly inferior position of women in today's labor market, especially those of mature age. Women tend to be paid less and have more difficult access to top positions. Women are at much greater risk of dropping out of the labor market than men due to the collision of work and family life. The percentage of women's employment clearly decreases with the arrival of more children. Generally, these women return to the labor market; however, the break in their professional activity usually lasts up to a dozen years, which makes women appreciate work and the opportunity for professional development more than men. There is also the issue of stereotypes and social and media pressure that a woman's role is to run the home and take care of children and elderly family members, and work and career is a secondary issue that should not interfere with these responsibilities [100]. As can be seen, this kind of tendency is not evident among men.

Based on the above analyses, it can be concluded that representatives of the "silver" generation are characterized by an instrumental attitude towards work, which they treat mainly as a means to satisfy their basic needs, such as prosperity, security, financial stability, and respect. Less popular is the autotelic attitude, where work is an end in itself and a way to satisfy the need for self-development. Suggesting positive information for employers, quite low marks were given by the respondents to the statement characterized by a punitive attitude, i.e., perceiving work as a punishment, something unpleasant that should be avoided. This may indicate their rather high work ethos, especially among

women. Undoubtedly, work occupies an important place in the lives of representatives of the "silver" generation; however, private life, i.e., the opportunity to spend time with family and friends, seems to be more important to them. This is especially important in the context of duties related to helping raise children (grandchildren) and taking care of seniors. Therefore, a fundamental issue for employees of the 50+ generation is the ability to manage their working time in a balanced and flexible way.

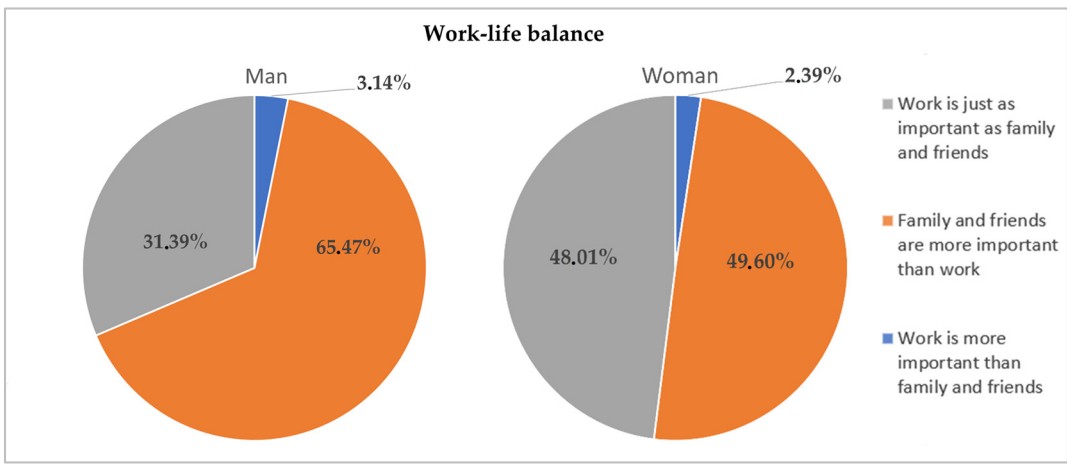

**Figure 6.** Work–life balance importance.

### 3.1.3. Professional Satisfaction of the "Silver" Generation

For the research hypothesis H$_3$ which says that:

*There are significant differences between the level of professional satisfaction of women and men representing the employees of the "silver" generation.*

There are no grounds to reject the null hypothesis that there are no differences between the groups (Z = −1.19; *p* = 0.23594). The answers given by women do not differ significantly from the answers given by men.

In order to investigate professional satisfaction of representatives of the "silver" generation, respondents were asked to assess three indicators (professional career development, professional fulfillment, and current job satisfaction) and to assign ratings to each of them on a scale of 1 to 10, where 1 meant "I am very dissatisfied" and 10 meant "I am very satisfied". Together, these indicators formed the professional satisfaction index. These ratings are presented in Table 5 as an arithmetic average of the responses obtained.

**Table 5.** Professional satisfaction index.

|  | **Woman** | **Man** | **Average Value** |
|---|---|---|---|
| Current job satisfaction | 7.24 | 7.07 | 7.18 |
| Professional fulfillment | 7.20 | 7.04 | 7.14 |
| Career development | 6.98 | 6.83 | 6.92 |
| Professional satisfaction index | 7.14 | 6.98 | 7.08 |

According to research, the professional satisfaction index of representatives of the "silver" generation is 7.08 and has very similar values for both women (7.14) and men (6.98). Taking into account the values taken separately by each of the examined indicators, it should be stated that they are also at a similar level; respectively, they are 7.18 for the current job satisfaction, 7.14 for professional fulfillment, and 6.92 for career development. It should be noted that the ratings given by women did not differ significantly from those given by men.

The above results show that the representatives of the "silver" generation surveyed assess the level of their professional satisfaction quite highly. They feel rather professionally fulfilled and they are satisfied with their current job and the career point they have reached there.

### 3.1.4. "Silver" Generation Professional Career Development

For the research hypothesis $H_4$ which says that:

*There are significant differences between the expectations regarding the professional career development of women and men representing the employees of the "silver" generation.*

The null hypothesis of no differences between the groups was rejected in favor of the alternative hypothesis that the differences between the groups are significant (Z = −2.67; *p* = 0.00769). The answers given by women differ significantly from the answers given by men. Women gave, on average, higher marks in questions concerning the prospects of their professional career development.

To determine the professional career prospects of employees of the "silver" generation (50+), respondents were asked to evaluate indicators that characterize the conditions and factors that motivate them to continue working after reaching retirement age and to assign ratings to each of them on a scale of 1 to 10, where 1 represents the least significant values and 10 the most significant. These indicators have been formulated in accordance with the assumptions of Edgar Schein's "career anchors" theory [84,85].

A fundamental issue from the point of view of the functioning of the labor market is to determine whether employees of the "silver" generation want to work after reaching retirement age (Figure 7). According to the conducted research, the vast majority of respondents (82% of women and 80% of men) are willing to continue working after reaching retirement age but would prefer a limited-time job (47% of women and 45% of men, respectively). Factors that constitute the greatest motivation to continue working after reaching retirement age (Table 6) included a good work atmosphere (8.72), high salary (8.58), stable employment conditions (8.54), and low stress level (8.39). The factors that received the lowest scores were foreign trips, internships and trainings (5.42), and the possibility of promotion and development (6.71). Women scored higher than men by about 0.6 points.

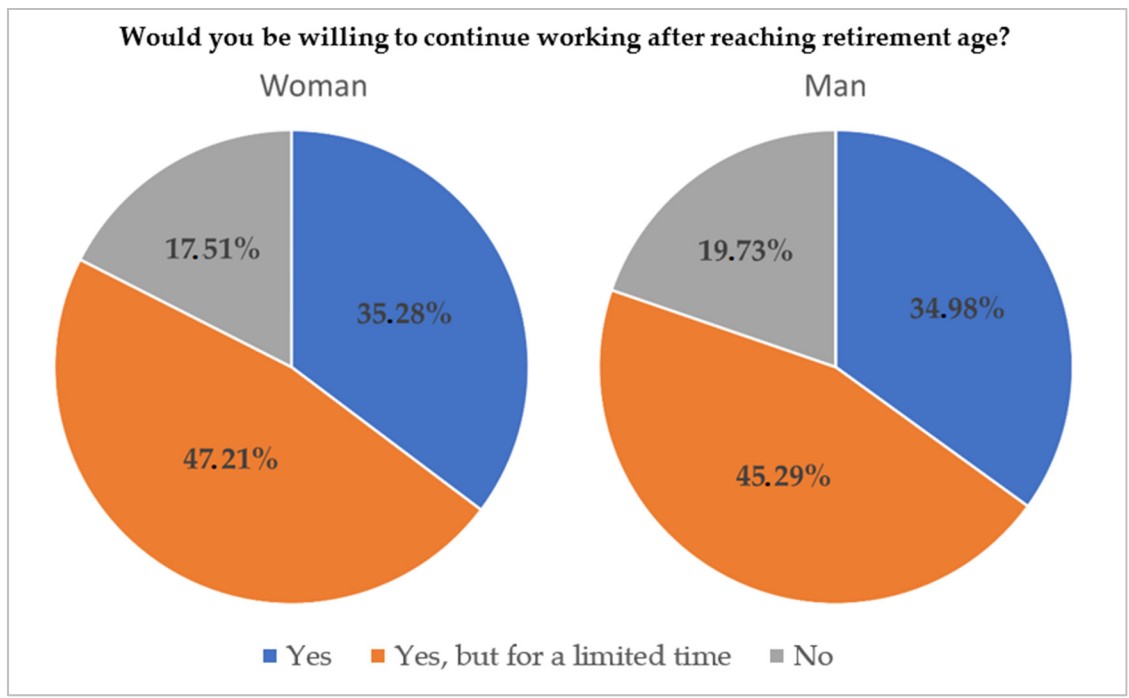

**Figure 7.** Willing to continue working after reaching retirement age.

**Table 6.** Factors that motivate workers to continue working after reaching retirement age.

|  | Woman | Man | Average Value |
|---|---|---|---|
| Good atmosphere at work | 8.94 | 8.35 | 8.72 |
| High earnings | 8.84 | 8.12 | 8.58 |
| Permanent employment contract, company stability | 8.79 | 8.11 | 8.54 |
| Low stress levels | 8.64 | 7.95 | 8.39 |
| Flexible time and form of work | 8.21 | 7.53 | 7.96 |
| Opportunity to pursue your interests | 7.45 | 7.16 | 7.34 |
| Possibility of promotion and development | 6.91 | 6.36 | 6.71 |
| Foreign trips, internships and trainings | 5.42 | 5.42 | 5.42 |
| Nothing will convince me | 0.02 | 0.03 | 0.02 |

The next step aimed at outlining the prospects for the career development of employees of the "silver" generation was to examine what factors and circumstances they identify with the further development of their careers (Figure 8). Respondents most strongly identify their professional future with high salary (8.49), job satisfaction (8.25), and work–life balance (8.20). The circumstances with the lowest scores were new challenges (6.98) and promotion (6.51). The ratings presented in Table 6 and Figure 8 are an arithmetic average of the responses obtained.

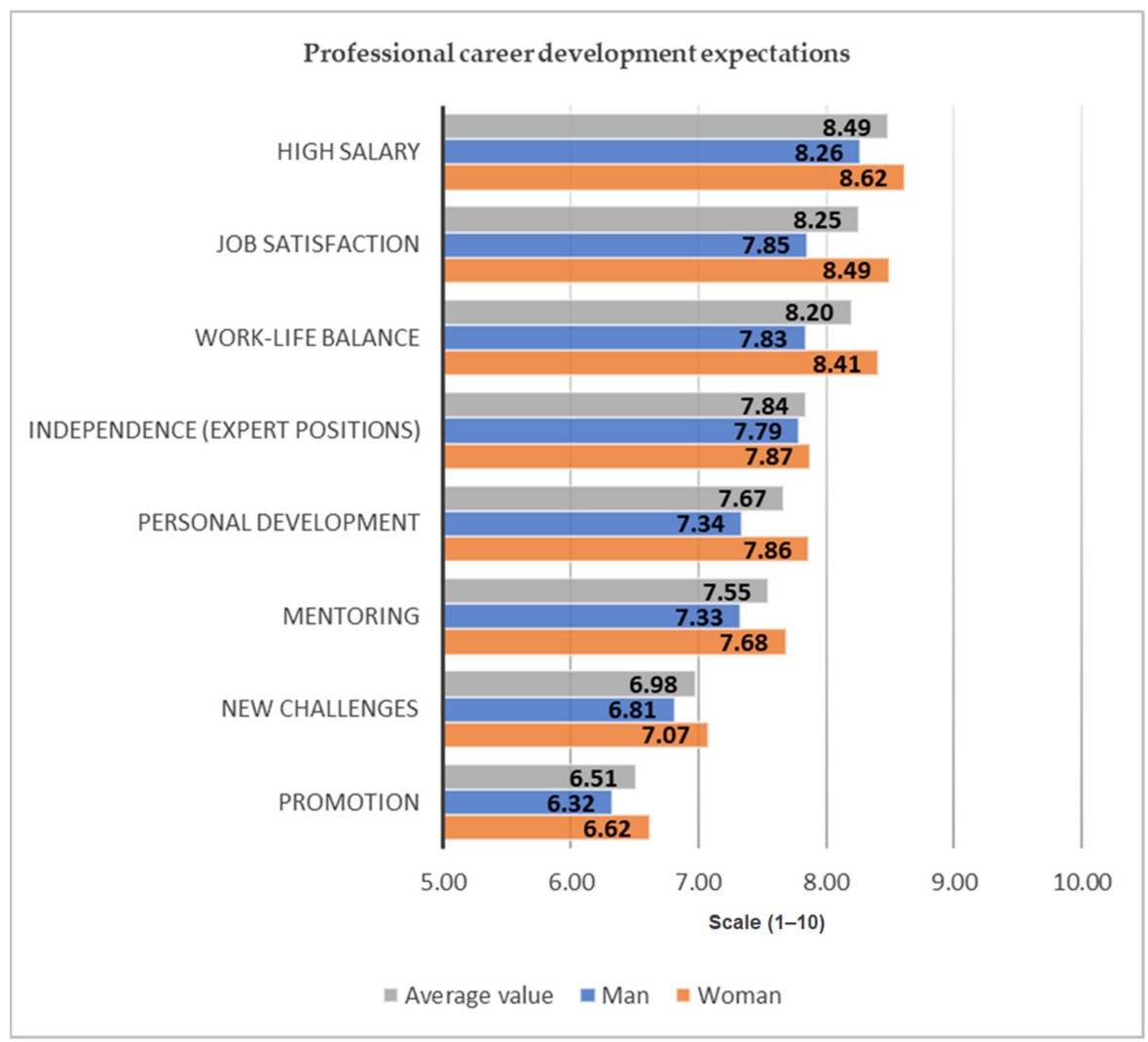

**Figure 8.** Professional career development expectations.

In order to clarify the characteristics presented above, the respondents were additionally asked to answer two questions concerning the issue of professional mobility and mentoring. In both cases, the responses given by women differed from those given by men. Research has shown that only 22% of surveyed men and 18% of women are ready to go abroad for a longer period of time (over a year) in order to take up a well-paid position, 25% of respondents are undecided, and as many as 46% of them do not see such a possibility (Figure 9). Women's greater reluctance than men to travel abroad for longer periods of time for work is likely due to their heavy burden of additional responsibilities associated with raising children (grandchildren) and caring for elderly family members.

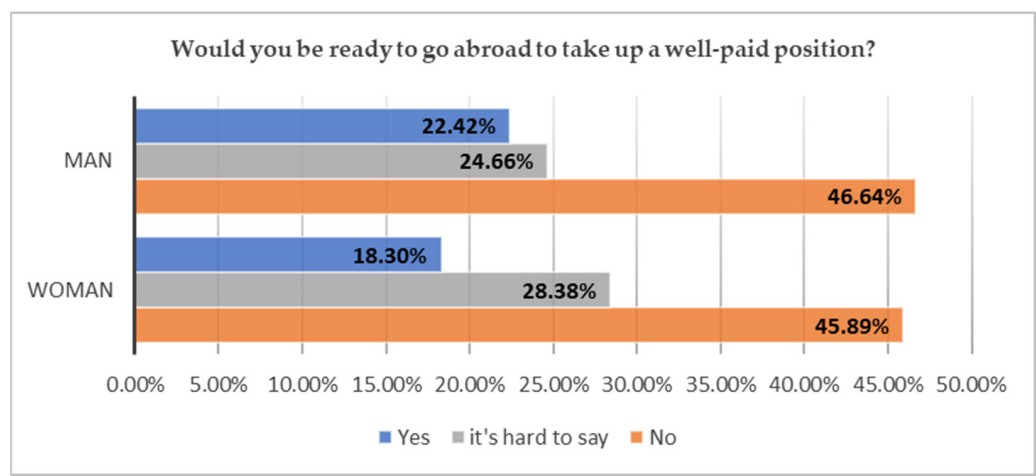

**Figure 9.** Preparedness to go abroad to take up a well-paid position.

Readiness to share experience and actively participate in training young employees for similar positions is expressed by over 55% of surveyed men and 58% of women, while 24% of men and only 14% of women are against such initiatives (Figure 10). Interestingly, up to 27% of women and 20% of men believe that they do not have sufficient competencies.

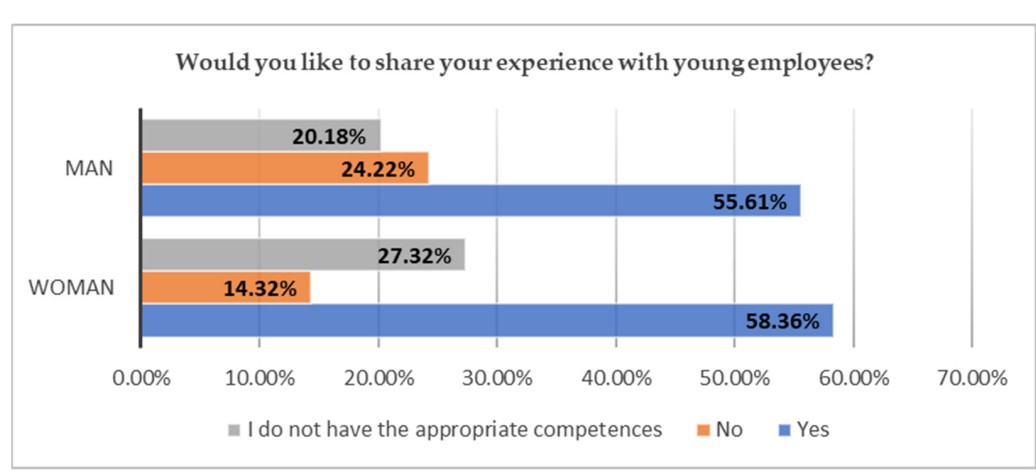

**Figure 10.** Readiness to share experience with young employees.

In summary, it should be stated that representatives of the "silver" generation (50+) are ready to continue working after reaching retirement age, which is very good news for the Industry 5.0 labor market. The factors that have a fundamental impact on decisions to delay retirement and remain professionally active are employment stability, financial security, low stress levels, and a good atmosphere at work. Equally important for representatives of the "silver" generation is also the possibility of dedicating more time to family and friends,

which is manifested by the willingness to work for a limited time. Contrary to stereotypical opinions about "Generation X", the surveyed employees are not workaholics [101] and highly value the possibility of maintaining a balance between private and professional life. They identify their professional future with stability and security rather than with dynamic development, which is manifested by reluctance to all kinds of change, such as mobility related to work. On the other hand, employees of the "silver" generation express their willingness to share their experiences with younger employees, which should be an additional impulse to build multigenerational teams.

### 3.2. Ordinal Logistic Regression

To verify the main research hypothesis ($H_G$) that says that:

*Characteristics such as the hierarchy of values, attitudes towards work, and professional satisfaction of both men and women employees of the "silver" generation have a significant, positive impact on the further development of their professional careers.*

Within the group of men and (separately) women, ordinal logistic regression models were built. The following partial hypotheses were formulated for each model:

($H_{Gm}$): *Features such as the hierarchy of values, attitudes towards work, and professional satisfaction of men have a significant, positive impact on the further development of their professional careers;*

($H_{Gw}$): *Features such as the hierarchy of values, attitudes towards work, and professional satisfaction of women have a significant, positive impact on the further development of their professional careers.*

#### 3.2.1. Model I (Men)

Dependent variable: IV—career development.

Independent variables: I—hierarchy of values, II—attitude towards work, and III—professional satisfaction.

After conducting the log-rank test I (Table 7), it was decided that the model would include the independent variables I and II.

**Table 7.** Log-rank test results for model I.

|  | Degrees | Log Rank | $\chi^2$ | $p$ |
|---|---|---|---|---|
| Free expression | 9 | −425.215 |  |  |
| I. Hierarchy of values | 1 | −389.156 | 72.12 | 0.000000 |
| II. Attitudes towards work | 1 | −376.811 | 24.69 | 0.000001 |
| III. Professional satisfaction | 1 | −376.797 | 0.03 | 0.866300 |

The value of the scaled $\chi^2$ coefficient is 2.68, indicating excessive scatter in the data. Hence, a model adjustment was carried out by estimating the scatter. Based on the analysis of Pearson's plot of residuals, it was found that case No. 473 significantly deviates from the others, justifying its omission from further analysis. The estimated scale parameter after adjustment was 1.00 (Figure 11).

After running the logistic regression model and the Wald test, the odds ratios obtained (Table 8) show that a unit increase in variable I increases the probability of accepting higher levels of variable IV by 3%; thus, the chance of men being interested in further professional career increases. Similarly, with a unit increase in variable II, the probability of adopting higher levels of variable IV increases by 4%, i.e., the chance of men being interested in further professional career increases.

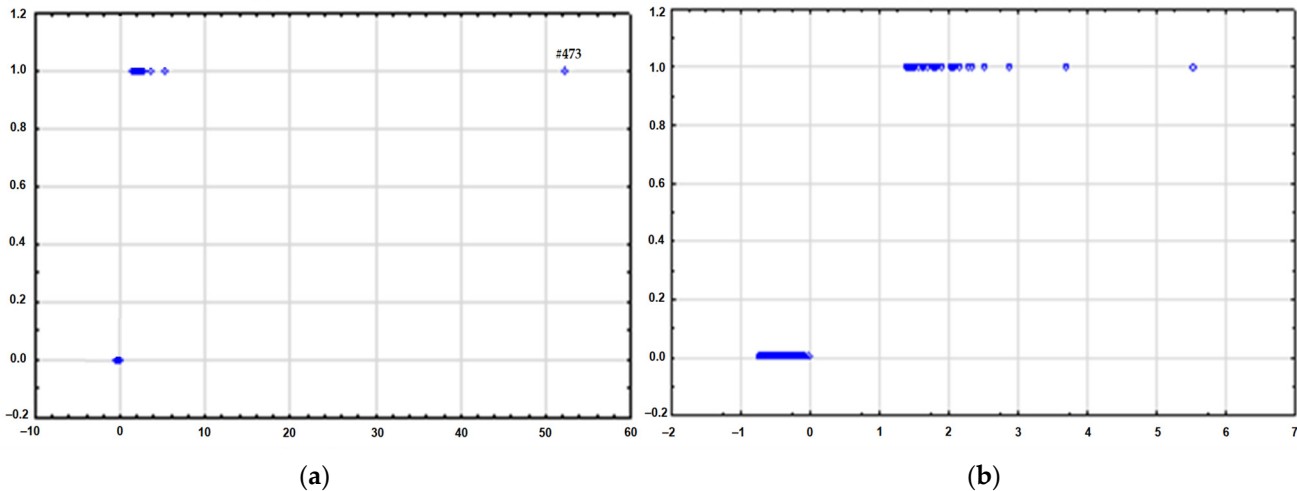

**Figure 11.** Pearson residual plots before (**a**) and after (**b**) removing case #473 for class 9.

**Table 8.** Results of ordinal logistic regression for model I (male).

|  | Rating | OR | Error Std. | Stat. Wald | GU Top 95% | GU Bottom 95% | *p* |
|---|---|---|---|---|---|---|---|
| I. Hierarchy of values | −0.02817 | 0.97 | 0.011001 | 6.56 | −0.04973 | −0.0066 | 0.010457 |
| II. Attitudes towards work | −0.04502 | 0.96 | 0.010098 | 19.88 | −0.06481 | −0.02523 | 0.000008 |

These results allow for partial confirmation of the main research hypothesis:

$H_{Gm}$—*the hierarchy of values and attitudes towards the work of <u>men</u> from the "silver" generation have a significant, positive impact on the further development of their professional careers.*

Men from the "silver" generation with a more polarized system of values and more established attitudes toward work can be assumed to be more interested in further development of their professional career. On the other hand, the level of professional satisfaction of men from the "silver" generation does not affect their interest in further development of their professional career.

3.2.2. Model II (Women)

Dependent variable: IV—career development.

Independent variables: I—hierarchy of values, II—attitude towards work, III—professional satisfaction.

The results of the log-rank I test (Table 9) are significant for all independent variables, therefore all three variables are included in the model.

**Table 9.** Log-rank I test results for model II (women).

|  | Degrees | Log-Rank | $\chi^2$ | *p* |
|---|---|---|---|---|
| Free expression | 8 | −694.638 |  |  |
| I. Hierarchy of values | 1 | −650.325 | 88.63 | 0.000000 |
| II. Attitudes towards work | 1 | −640.609 | 19.43 | 0.000010 |
| III. Professional satisfaction | 1 | −637.075 | 7.07 | 0.007844 |

The value of the scaled coefficient $\chi^2$ is 1.17, which means that there is no excessive dispersion of the data. Based on the visual evaluation of the Pearson residual plots, it was found that no cases need to be skipped. The logistic regression model and the Wald test (Table 10) allowed the evaluation of model II parameters (odds ratio, standard error, 95% ranges for evaluation).

**Table 10.** Results of ordinal logistic regression for model II (women).

|  | Rating | OR | Error Std. | Stat. Wald | GU Top 95% | GU Bottom 95% | *p* |
|---|---|---|---|---|---|---|---|
| I. Hierarchy of values | −0.02746 | 0.97 | 0.007591 | 13.09 | −0.04234 | −0.01258 | 0.000298 |
| II. Attitudes towards work | −0.03227 | 0.97 | 0.006755 | 22.82 | −0.04551 | −0.01903 | 0.000002 |
| III. Professional satisfaction | 0.041154 | 1.04 | 0.016484 | 6.23 | 0.008846 | 0.07346 | 0.012539 |

In the case of model II (women), the obtained odds ratios (Table 10) show that a unit increase in variable I increases the probability of accepting higher levels of the variable described as IV by 3%; thus, the chance of women becoming interested in further professional careers increases. Similarly, a unit increase in variable II increases the probability of adopting higher levels of variable IV by 3%; thus, the chance of women becoming interested in further professional careers increases. Interestingly, a unit increase in variable III decreases the probability of adopting higher levels of variable IV by 4%; thus, the chance of women being interested in further professional careers decreases.

These results partially confirm the main research hypothesis:

H$_{Gw}$: *the hierarchy of values and attitudes toward the work of women of the "silver" generation have a significant positive impact on the further development of their professional careers. Professional satisfaction of women of the "silver" generation has a significant negative impact on the further development of their professional careers.*

It can be assumed that the "silver" generation women with a more polarized system of values and more established attitudes toward work are more interested in further development of their professional careers. On the other hand, the "silver" generation women, who are more professionally fulfilled, are less interested in further development of their professional career. This is probably due to the fact that professionally fulfilled women with adequate financial security are more likely to decide to retire and engage in helping their immediate family (e.g., in raising grandchildren).

### 3.3. Implications for Sustainable Human Capital Management in Enterprises

The rapidly progressing decline in the number of people of working age in the countries of the European Union, combined with the need to maintain the production and service capacity of enterprises, implies the need to extend the working time of employees. To achieve this, it is necessary to develop solutions that will encourage older workers to postpone the decision to retire. The research results presented in this article allowed for the formulation of the following guidelines, which are also implications for the sustainable management of human capital in the era of Industry 5.0:

- Due to the different expectations of women and men, the detailed conditions under which the employment contract of a 50+ employee is to be extended should be agreed upon individually with the employee;
- In order to maintain the desired balance between private and professional life, it is necessary to enable interested employees aged 50+ to continue employment on a limited-time basis or to adopt a different, flexible form of working time organization (e.g., hybrid work);
- In order to build the sense of security of employees aged 50+, they should be provided with stable employment conditions, which include fair remuneration related to their professional experience, financial bonuses depending on work results, and non-wage benefits, such as additional insurance and a medical package;
- To reduce the level of stress and to create the right atmosphere at work, care should be taken to ensure equal, non-discriminatory treatment of employees;
- In order to ensure partner relations at work, changes affecting the place, conditions, and nature of work should be introduced only after individual consultations with the employee;

In order to manage the human capital of enterprises and institutions more sustainably and effectively, the organization of work should be based on multigenerational teams, which will enable the transfer of professional experience, competencies, and skills between older and younger employees (mentoring and reverse mentoring).

## 4. Conclusions

In summary of the above research, it should be stated that representatives of the "silver" generation are conservatives focused on family and private life, who value personal security, health, stabilization, and compliance. They consider universal values, such as family, honesty, and love and happiness, to be the most important in life, which they place much higher than career and professional position. Representatives of the "silver" generation are characterized by an instrumental attitude to work, which they treat mainly as a means to satisfy their basic needs, such as prosperity, security, financial stability, and respect. Undoubtedly, work occupies an important place in the lives of representatives of the "silver" generation; however, private life, that is, the opportunity to spend time with family and friends, seems to be more important to them. These people express their readiness to continue working after reaching retirement age, which is very good news from the point of view of the needs of the modern labor market. Factors that may motivate them to work longer include employment stability, financial security, the ability to maintain a balance between work and private life, as well as a low level of stress and a good atmosphere at work. Representatives of the "silver" generation identify their professional future with stability and security rather than dynamic development, manifested by a reluctance to change at work, such as long business trips. On the other hand, employees of the "silver" generation are ready to share their experiences with younger employees, which should be an impulse to build multigenerational teams.

In the course of the research work, it was confirmed that there are significant differences between male and female respondents in the evaluation of traits regarding their value hierarchy, attitudes toward work, and career prospects. Traits such as value hierarchy and attitudes toward work have a significant impact on the career development of both women and men of the "silver" generation, while professional satisfaction shows a significant (negative) impact on career development only for women.

In order to effectively utilize the potential inherent in the employees of the 50+ generation, it is necessary to implement innovative, customized solutions in the field of human capital management of the enterprise, such as, for example, implementing flexible forms of employment, maintaining partnership, inclusive and unbiased labor relations, and organizing work on the basis of multi-generational teams, formed for the purpose of performing a specific task. In addition, it is important to remember to treat modern technologies not only as a starting point and potential for increasing the efficiency of the enterprise but, above all, to apply a human-centered approach that puts basic human needs and interests at the center of the production process. In other words, instead of asking what can be done with new technology, ask what the technology can do for us. Instead of adapting workers' skills to the needs of a rapidly evolving technology, use that technology to adapt the production process to the worker's capabilities and needs.

It should be noted that these studies obviously have some limitations. The most important one seems to be the restriction of the research group to Polish citizens, whose peculiarities (history, social, and cultural characteristics) may in some way affect the results of the study. Therefore, it would be worth conducting similar research in other countries struggling with the problem of demographic decline (e.g., EU, USA, and Japan), which would allow for a more complete description of the "silver" generation at work. An interesting and needed direction for future research is also the issue of the functioning and management of multigenerational teams.

**Author Contributions:** Conceptualization, A.L.; methodology, A.L.; software, J.F.L.; validation, A.L. and J.F.L.; formal analysis, A.L.; investigation, A.L.; resources, A.L.; data curation, J.F.L.; writing—original draft preparation, A.L.; writing—review and editing, A.L. and J.F.L.; visualization, J.F.L.;

supervision, A.L.; project administration, A.L.; funding acquisition, A.L. and J.F.L. All authors have read and agreed to the published version of the manuscript.

**Funding:** This research received no external funding.

**Institutional Review Board Statement:** Ethical review and approval were waived for this study due to the non-interventional questionnaire conducted in accordance with "ICC/ESOMAR International Code on Market, Opinion and Social Research and Data Analytics" (https://iccwbo.org/publication/iccesomar-international-code-market-opinion-social-research-data-analytics/ accessed on 10 January 2022). All participants were informed that anonymity was assured, why the research was being conducted, how their data would be used, and that there were no risks associated with their participation in the study.

**Informed Consent Statement:** Informed consent was obtained from all subjects involved in the study.

**Data Availability Statement:** The data presented in this study are available on request from the corresponding author.

**Conflicts of Interest:** The authors declare no conflict of interest.

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
