# Peer review of "“Silver” Generation at Work—Implications for Sustainable Human Capital Management in the Industry 5.0 Era"

_sustainability, doi:10.3390/su15010194_

Round 1

Reviewer 1 Report

Please explain in more detail why you chose this particular case indicated in fig. 13. What are its main conclusions?

What do you think are the reasons for such drastic differences in attitudes between men and women wishing to realize themselves abroad?

Author Response

We would like to thank the reviewers for their careful comments and constructive suggestions to our submission. In the revision, we have taken the reviewers' views very seriously, and made every possible effort to address all concerns. The paper has been duly revised, and the replies to the comments can be found in the attachment.

Reviewer 2 Report

This is a very extensive article, presenting detailedresearch results.it seems that a reference to the title would increase the value of the article. The title is about sustainable human capital management. there is little information about sustainable capital management and human and social capitalin the literature review. in addition, the conclusions could be a bit more elaborate. However, the use of the term industry 5,0 in the title is something to think about. In the Industry 5,0 concept, the human factor regains its rightful position in the center of the production process. Industry 5,0 is looking for solutions that enable a harmonious and synergistic combination ofthe work of people and devices. Nevertheless, for the authors of the article,it is only a background. Regardless of the comments,the article is valuable and worth reading. 

Author Response

(The authors gave the same response as above.)

Reviewer 3 Report

Review “SILVER” GENERATION AT WORK - IMPLICATIONS FOR SUSTAINABLE HUMAN CAPITAL MANAGEMENT IN THE INDUSTRY 5.0 ERA

The theme of the article seems very interesting to me, I am going to make some observations about this article

1-The introduction is very brief, it should be expanded by commenting, for example, the importance of the theme, the framework of analysis, etc.

2- I think it is necessary to restructure section 2 Literature review. It is excessively separated; it should be given a greater continuity.

3-I think that section 2 Literature review would be better structured if subsection 2.1 Generational cohorts were moved to the introduction

4-Subsection 2.3 Theoretical background and research hypotheses should go to the introduction: it is more appropriate to put the hypotheses in the introduction than in the literature review.

5-Section 4.2 Data analysis (line 288) is actually Section 3.2

6-More information is needed on the basic work survey. Please, it is necessary to provide information about the type of sampling carried out, the reliability of the survey, the type of questions asked, etc.

7- If the questions in the questionnaire are of ordinal scale, it does not make sense to make a normality test. If this is the case, the use of the descriptors employed is quite questionable..

8-It is necessary to say how the means of figure 2 and following have been calculated (weighted or not)

9-I find the results shown in figure 5 strange, since, in general terms, it is women who stay at home to care for sick relatives or are promoted less professionally to dedicate themselves to their family

10- The graphs are very repetitive; the authors could change the type of graphs and replace them in some cases with tables.

11-Summarize 4.1.2, is very repetitive.

12. The HG hypothesis, "the hierarchy of values, attitudes towards work, and professional satisfaction of men and women representing employees of the “silver” generation have a significant impact on the further development of their professional careers” is quite obvious and has no interest to contrast it.

13. The conclusions are very brief; I think they should be expanded by introducing some additional discussion

Author Response

We would like to thank the reviewers for their careful comments and constructive suggestions to our submission. In the revision, we have taken the reviewers' views very seriously, and made every possible effort to address all concerns. The paper has been duly revised and the replies to the comments can be found in the attachment.

Round 2

Reviewer 3 Report

Suggested changes have been taken into consideration. In my opinion the article can be published